

# Lahar events in the last 2,000 years from Vesuvius eruptions. Part 2: Formulation and validation of a computational model based on a shallow layer approach

Mattia de' Michieli Vitturi[1], Antonio Costa[2], Mauro A. Di Vito[3], Laura Sandri[2], Domenico M. Doronzo[3]

[1]Istituto Nazionale di Geofisica e Vulcanologia, Sezione di Pisa, Pisa, 56125, Italy

[2]Istituto Nazionale di Geofisica e Vulcanologia, Sezione di Bologna, Bologna, 40128, Italy

[3]Istituto Nazionale di Geofisica e Vulcanologia, Osservatorio Vesuviano, Napoli, 80124, Italy

*Correspondence to*: Mattia de' Michieli Vitturi (mattia.demichielivitturi@ingv.it)

**Abstract.** In this paper we present a new model for the simulation of lahars, based on the depth-averaged code IMEX-SfloW2D with new governing and constitutive equations introduced to better describe the dynamics of lahars. A thorough sensitivity analysis is carried out to identify the critical processes (such as erosion and deposition) and parameters (both numerical and physical) controlling lahar runout, using both synthetic and real cases topographies. In particular, an application of the model to a syn-eruptive lahar from a reference-size eruption from Somma-Vesuvius, affecting the Campanian Plain (Southern Italy), described in Di Vito et al. (this issue), is used in this work for the sensitivity analysis. Effects of erosion and deposition are investigated by comparing simulations with and without these processes. By comparing flow thickness and area covered by the flow and their evolution with time, we show that the modelling of both the processes is important to properly simulate the effects of the bulking and debulking and the associated changes in rheology. From a computational point of view, the comparison of simulations obtained for different numerical grids (from 25 m to 100 m), scheme order, and grain size discretization were useful to find a good compromise between resolution and computational speed. The companion paper by Sandri et al. (this issue) shows an application of the presented model for probabilistic volcanic hazard assessment for lahars from Vesuvius deposits in the Neapolitan area.

## 1 Introduction

Water saturated flows made from volcanic deposits are known as "lahar", an Indonesian term used to indicate muddy flows. As typical in the volcanological literature, here we will use the term lahar to denote any water saturated flows, from hyperconcentrated flow, carrying up to 50 vol % sediment, to lower concentration flows (< 10 % sediment). These wet granular flows are commonly characterised by a high flow density and can have high flow velocity, generating large dynamic pressures able to destroy even buildings and infrastructures. Moreover, this kind of flows can inundate large areas, disrupting ground transportation networks, human settlements, power lines, industry, agriculture (e.g., Zanchetta et al., 2004).

Lahars can form from the remobilization of unconsolidated tephra, as for the hundreds of lahars generated by torrential rains after the 1991 Pinatubo eruptions in the Philippines (Van Westen and Daag, 2005). In other cases, as at Mount St. Helens, lahars can result from dome collapses and the associated volcanic explosions (Scott, 1988). Additionally,



devastating lahars can form when a pyroclastic flow melts snow or ice caps (Major and Newhall, 1989), as occurred for
the 1995 eruption on the glaciated Nevado del Ruiz, Colombia (Pierson et al., 1990). Mt. Rainier is another example of
volcano that experienced several lahars of this kind in the past. Lahars can form also in eruptions beneath crater lakes, as
at Keluth, Indonesia (Mastin and Witter, 2000) and Ruapehu, New Zealand (Lecointre et al., 2004).
If lahars are generated before, during, or after the eruption they are named pre-eruptive, syn-eruptive, or post-eruptive
lahars (Vallance and Iverson, 1995). The term syn-eruptive must not be taken literally, but indicates a lahar generated
during or in the period immediately following an eruption. Besides a triggering mechanism, generation of a lahar requires
*i*) an adequate water source, which can be hydrothermal water, rapidly melted snow and ice, crater lake water, and rainfall
runoff, *ii*) abundant unconsolidated debris that typically includes pyroclastic flow and fall deposits, glacial drift,
colluvium, and soil, and *iii*) steep slopes and substantial relief at the source (IAEA, 2016). Because lahars are water
saturated flows, for which both liquid and solid interactions are fundamental, their behaviour is different from other
related phenomena common to volcanoes such as debris avalanches and floods. In terms of fragment size distribution, the
material carried by lahars ranges in diameter from about $10^{-6}$ m to 10 m. Lahars can have temperature up to 100 °C and
can change character downstream, through processes of flow bulking (erosion and incorporation of secondary debris as
they move downstream) and debulking (a process in which the lahar selectively deposits certain particles, owing to their
size or density, as it moves downstream). Primary particles in lahar deposits derive from contemporaneous eruption
deposits or, in the case of avalanche induced lahar deposits, from the original avalanche mass; secondary particles derive
from the erosion and incorporation of downstream volcaniclastic debris, alluvium, colluvium glacial drift, bedrock, etc.
Many properties of lahars including, but not limited to, particle concentration, granulometry and componentry, bulk
rheology and velocity are highly variable in both time (i.e. unsteadiness) and space (i.e. non-uniformity).
Several methods have been proposed to assess the related hazard, ranging from simple empirical models like LAHARZ
(Iverson et al., 1998), which can be used to estimate the inundated areas, to geophysical mass-flow models which use
different rheological laws, such as Newtonian, Bingham, Bagnold, or Coulomb models, depending on flow behaviour,
(e.g., TITAN2D, Pitman et al., 2003; Patra et al., 2005; FLO2D, O'Brien et al., 1993; VolcFlow, Kelfoun and Druitt,
2005; Kelfoun et al., 2009) and can furnish values of critical variables, such as velocity and dynamic pressure. A different
approach, based on a fully three-dimensional model of two-phase flows, can be found in Dartevelle (2004) and Meruane
et al. (2010). One of the most general two-phase debris-flow models was developed by Pudasaini (2012), and it includes
many essential physical phenomena observable in debris flows. Mohr-Coulomb plasticity is used to close the solid stress.
The reader is addressed to Pudasaini (2012) and references therein for a general review of the topic. More recently,
building on the Pudasaini (2012) two-phase flow model, Pudasaini and Margili (2019) presented a new mass flow model
(r.avaflow, https://www.landslidemodels.org/r.avaflow) accounting for the complexity of geomorphic mass flows
consisting of coarse particles, fine particles, and viscous fluid.
In this work we present a new simplified model developed for the aim of lahar hazard assessment. The model, discussed
in Section 2, is based on the Saint-Venant depth-averaged equations, coupled with source terms accounting for friction
and with terms for erosion/deposition of solid particles. Then in Section 3 we present a few examples of model validation
and applications, and in Section 4 a short discussion and conclusion.
**2 Physical-numerical model**



The physical model for lahars is based on the shallow layer approach and on the solutions of a set of depth-averaged
transport equations. As we explain below numerical solution was obtained by modifying the IMEX-SfloW2D code (de'
Michieli-Vitturi et al., 2019), with new governing and constitutive equations introduced to better simulate lahars
dynamics. In this section, we briefly introduce all model variables, and we describe the governing equations.

**2.1 Model governing equations**

**2.1.1 Depth-averaged transport equations**

In this section, we present the set of partial differential equations governing the dynamics of lahars. Assuming that the
lahar flow is a homogeneous mixture of water and $n_s$ solid phases (see Fig. 1), its density $\rho_m(x, y, t)$ is defined in terms
of the volumetric fractions $\alpha_{(\cdot)}$ and densities $\rho_{(\cdot)}$ of the components:

$$\rho_m = \alpha_w \rho_w + \sum_{i_s=1}^{n_s} \alpha_{s,i_s} \rho_{s,i_s} \tag{1}$$

where the subscript $w$ denotes the water phase and the subscript $s, i_s$ denote the class $i_s$ of the solid phase. Equations are
written in global Cartesian coordinates $(x, y)$, with $x$ and $y$ orthogonal to the $z$−axis, assumed to be parallel to
gravitational acceleration $g = (0,0,g)$. We denote the flow thickness with $h(x, y, t)$ and the depth-averaged horizontal
components of the flow velocity with $u(x, y, t)$ and $v(x, y, t)$, assuming that, due to the flow turbulence, solid phases are
well mixed with the liquid carrier phases, and they have the same horizontal velocity. The flow moves over a topography,
described by the variable $B(x, y, t)$. In principle, topography can change with time, but as first approximation we neglect
the changes associated with erosion and deposition, while these processes are modelled and accounted for the flow
dynamics. Thus, we assume the topography being a function of space only.
With the notation introduced above, conservation of mass for the flow mixture writes in the following way:

$$\frac{\partial \rho_m h}{\partial t} + \frac{\partial (\rho_m h u)}{\partial x} + \frac{\partial (\rho_m h v)}{\partial y} = \sum_{i_s=1}^{n_s} [\rho_{s,i_s}(E_{s,i_s} - D_{s,i_s})] + \rho_w \left\{ D_w + \frac{\alpha_d}{1 - \alpha_d} \sum_{i_s=1}^{n_s} [(E_{s,i_s} - D_{s,i_s})] \right\}, \tag{2}$$

where $E_s$ and $D_s$ are the volumetric rate of erosion and deposition of solid particles, respectively, and $D_w$ is the rate of
loss of water, not associated with the deposition of particles (for example associated with evaporation or other processes).
The first term on the right-hand side accounts for the loss and entrainment of solid particles, while the last term accounts
for the loss of water. This term accounts not only for the loss due to the rate $D_w$, but also for the loss associated with
particle erosion and sedimentation. In fact, we assume both the pre-existing erodible layer and the flow deposit water-
saturated, with the volume fraction of water given by $\alpha_w$.
The two equations for momentum conservation are:



$$\frac{\partial(\rho_m hu)}{\partial t} + \frac{\partial}{\partial x}\left(\rho_m hu^2 + \frac{1}{2}\rho_m gh^2\right) + \frac{\partial}{\partial y}(\rho_m huv) = -\rho_m gh\frac{\partial B}{\partial x} + F_x$$
$$-u\left[\sum_{i_s=1}^{n_s}(\rho_{s,i_s}D_{s,i_s}) + \rho_w\left(D_w + \frac{\alpha_d}{1-\alpha_d}\sum_{i_s=1}^{n_s}D_{s,i_s}\right)\right],$$

(3a)


$$\frac{\partial(\rho_m hv)}{\partial t} + \frac{\partial}{\partial x}(\rho_m huv) + \frac{\partial}{\partial y}\left(\rho_m hv^2 + \frac{1}{2}\rho_m gh^2\right) =$$
$$-\rho_m gh\frac{\partial B}{\partial y} + F_y - v\left[\sum_{i_s=1}^{n_s}(\rho_{s,i_s}D_{s,i_s}) + \rho_w\left(D_w + \frac{\alpha_d}{1-\alpha_d}\sum_{i_s=1}^{n_s}D_{s,i_s}\right)\right],$$

(3b)

where $F = (F_x, F_y)$ is the vector of frictional forces and the last term on the right-hand side of both the equations considers
the loss of momentum associated with particle sedimentation. Please note that there are no terms associated with erosion
of solid particles in the momentum equations, because they do not carry any horizontal momentum within the flow,
although they change the inertia terms.
Flow temperature $T$ changes with entrainment of water and solid particles eroded from the underlying terrain, and this in
turn can change lahars property (for example viscosity). For this reason, we also solve for a transport equation for the
internal energy $e = C_v T$ (with $C_v$ being the mass averaged specific heat in the flow):

$$\frac{\partial}{\partial t}(\rho_m he) + \frac{\partial}{\partial x}(\rho_m hue) + \frac{\partial}{\partial y}(\rho_m hve)$$
$$= \sum_{i_s=1}^{n_s}\left[\rho_{s,i_s}C_{s,i_s}\left(T_s E_{s,i_s} - T D_{s,i_s}\right)\right] + \rho_w C_w \frac{\alpha_d}{1-\alpha_d}\sum_{i_s=1}^{n_s}\left[\left(T_s E_{s,i_s} - T D_{s,i_s}\right)\right]$$

(4)

where $C_s$, $C_w$ are the specific heats of solid and water, respectively, and $T_s$ is the substrate temperature before erosion. In
this equation, heat transfer by thermal conduction is neglected, as well as thermal radiation and heating due to friction.
Additional transport equations for the mass of $n_s$ solid classes are also considered:

$$\partial\frac{\left(s,i_s\rho_{s,i_s}h\right)}{\partial t} + \frac{\partial\left(\alpha_{s,i_s}\rho_{s,i_s}hu\right)}{\partial x} + \frac{\partial\left(\alpha_{s,i_s}\rho_{s,i_s}hv\right)}{\partial y} = \rho_{s,i_s}\left(E_{s,i_s} - D_{s,i_s}\right), i_s = 1, \dots, n_s$$

(5)

Finally, we have $n_s$ equations for the volume of solid particles in the water-saturated erodible layer:

$$\frac{\partial\alpha_{s,i_s}h_{s,i_s}}{\partial t} = \left(E_{s,i_s} - D_{s,i_s}\right), \quad i_s = 1, \dots, n_s$$

(6)

where $h_{s,i_s}$ is the thickness of each solid class in the layer, related to the total thickness $h_e$ of this layer by the
relationship:




$$h_e = \frac{1}{1 - \alpha_w} \sum_{i_s=1}^{n_s} h_{s,i_s}. \tag{7}$$

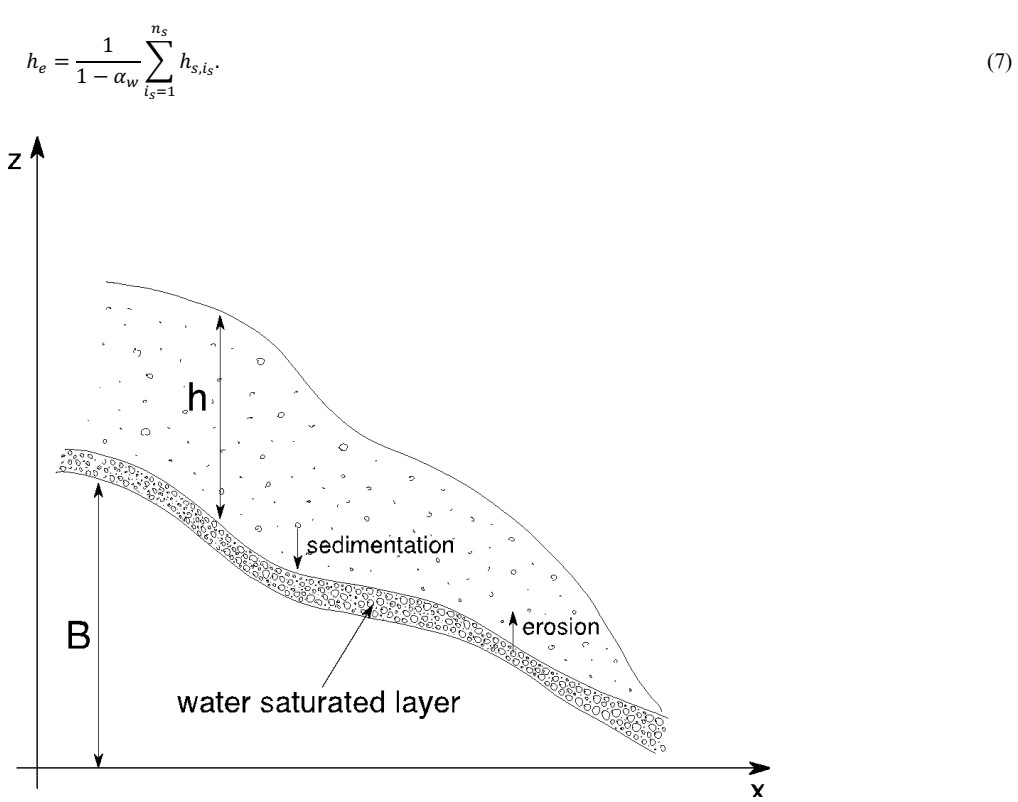

Figure 1. Sketch of the lahar.

### 2.1.2. Constitutive equations

The set of equations (1-7) constitute a set of $4 + n_s$ partial differential equations for the unknown state variables $Q = h$, $u, v, T, \alpha_{s,1}, \dots, \alpha_{s,n_s}$. In order to close the system and to be able to solve the equations, the terms accounting for friction, deposition, erosion should be defined as functions of the state variables $Q$.

The friction term appearing in the momentum equations is written in the following way:

$$F = (F_x, F_y) = \rho_m gh \left( \frac{u}{\sqrt{u^2 + v^2}} s_f, \frac{u}{\sqrt{u^2 + v^2}} s_f \right) \tag{8}$$

where $s_f$ is defined, accordingly to O'Brien et al. (1993), as the total friction slope, given by the sum of three non-dimensional terms:

$$s_f = s_y + s_v + s_t. \tag{9}$$



Here, $s_y$ is the velocity-independent yield slope, $s_v$ is the viscous slope and $s_t$ is the turbulent slope. These three terms,
as done in the numerical code FLO-2D, are written in the following way:

$$s_f = \frac{\tau_y}{\rho_m g h} + \frac{K\mu\sqrt{u^2 + v^2}}{8\rho_m g h^2} + \frac{n_{td}^2(u^2 + v^2)}{h^{4/3}} \tag{10}$$

where $\tau_y$ is yield strength, $K$ is an empirical resistance parameter, $\mu$ is fluid viscosity and $n_t$ is the Manning roughness
coefficient. In FLO-2D model (O'Brien et al., 1993), yield strength $\tau_y$ and fluid viscosity $\mu_m$ are defined through two
empirical relationships derived from field observations:

$$\mu_m = a_1 exp(b_1 \alpha_s) \tag{11a}$$


$$\tau_y = a_2 exp(b_2 \alpha_s) \tag{11b}$$


where $a_i$ and $b_i$ are coefficients defined by laboratory experiments and $\alpha_s$ is the total volumetric fraction of solid ($\alpha_s =$
$\sum_{i_s=1}^{n_s} \alpha_{s,i_s}$). In the original formulation of O'Brian et al., (1993) the empirical parameters $a_1$ e $b_1$ are model constants,
which do not vary with flow temperature. Here, we notice that the parameter $a_1$ has the units of a dynamic viscosity and
it can be seen as the limit viscosity of the mixture when the dispersed solid fraction goes to zero. Thus, it should represent
the dynamic viscosity of water. Commonly this parameter can be assumed to be constant, but in order to account for the
dependence of water viscosity on its temperature, which could potentially affect lahar dynamics and runout, here we
account for an additional correction factor $\Gamma(T^c)$, function of the temperature expressed in Celsius degrees:

$$a_1 = [\mu^{ref} \cdot \Gamma(T^c)]. \tag{12}$$


where $\mu^{ref}$ denotes the viscosity at a reference temperature $T^{ref}$. Following Crittenden et al. (2012), the equation used
to compute the factor $\Gamma(T^c)$ is given by:

$$\Gamma(T) = C \cdot \gamma \cdot 10^A$$


where:

$$\begin{cases} \gamma = 10^{-3}, & for\ 0 < T^c < 20°C \\ \gamma = (1.002 \cdot 10^{-3})(10^B), & for\ T^c \geq 20°C \end{cases}$$


$$\begin{cases} A = \dfrac{1301}{998.333 + 8.1855(T^c - 20) + 0.00585(T^c - 20)^2} - 1.30223 & for\ 0 < T^c < 20°C \\[3mm] A = \dfrac{1.3272(20 - T^c) - 0.001053(T^c - 20)^2}{T^c + 105} & for\ T^c \geq 20°C \end{cases}$$






and $C$ is a constant such that $\Gamma(T^{c,ref}) = 1$. With this choice, when $T^c = T^{c,ref}$ and $\alpha_s = 0$, $\mu = \mu^{ref}$. With respect to
the original work of O'Brian et al., (1993), also the original relationship for yield strength has been modified. In fact, here
we take:

$$\tau_y = a_2(exp(b_2\alpha_s) - 1) \tag{13}$$


In this way, yield stress disappears when solid fraction $\alpha_s$ goes to zero, recovering the Newtonian behaviour of water.

The values of the three components of the total friction slope (see Eq. 10) strongly depends on volumetric solid fraction,
on flow thickness and velocity. In Figure 2, for fixed values of the empirical parameters $a_i$ and $b_i$ ($i$=1,2) and for three
different values of the total solid volume fraction ($\alpha_s = 0$ on the left; $\alpha_s = 0.25$ in the middle; $\alpha_s = 0.5$ on the right), we
plotted the values of the three terms as function of flow thickness and velocity. These diagrams (in logarithmic scale for
all the variables) highlight how these terms can vary in a non-linear way by several orders of magnitude when thickness,
velocity and solid fraction vary in ranges that can be observed in lahars, potentially resulting in the presence of a stiff
term in the system of equations. For this reason, it is needed a robust solver that allows the coupling between the
gravitational and frictional terms to be accurately simulated.


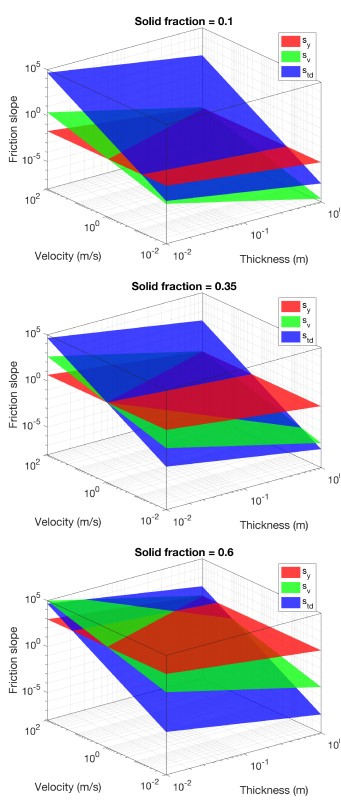



*Figure 2. Contribution of the yield slope ($s_y$), viscous slope ($s_v$) and turbulent slope ($s_{td}$) to the total friction slope for*
*three different solid volume fractions: 0% (left); 25% (middle); 50% (right). The friction parameters have the following*
*values: $K = 24.0$; $a_1 = 8.9 \cdot 10^{-4}$; $b_1 = 22.1$; $n_t = 0.1$; $a_2 = 0.272$; $b_2 = 22.0$.*
We also note that the presence of the yield strength term, i.e. a term independent of the velocity that opposes the motion,
allows the flow to stop with a thickness that depends on the slope of the topography and on the fraction of solid material
in the flow. This critical thickness can be calculated analytically and allows for the validation of the correct
implementation of the discretization of the friction terms in the numerical model. Below we present a figure illustrating
this relationship, where each line represents the critical thickness threshold line between the steady and unsteady condition
for different total solid percentage in the flow. We can see that, approximatively, an increase of 10% in the solid volume
fraction, for a fixed slope, corresponds to a factor 4.5 increase in the critical thickness. We also observe that such a critical
thickness is not only relevant for flow stoppage, but also for the initial triggering of the flow, and that this relationship
can be formulated also in terms of critical liquid volume fraction. Thus, given a thickness of the permeable layer and a
slope, we can compute the critical liquid volume fraction over which the lahar is triggered because the gravitational force
exceeds the yield strength. For example, a slope of 20° and a thickness of 1 m, a 60% liquid volume would trigger a lahar,
while a 50% liquid volume would not. It is also worth to note that these critical thresholds depend on the values of the
parameters for the yield strength.

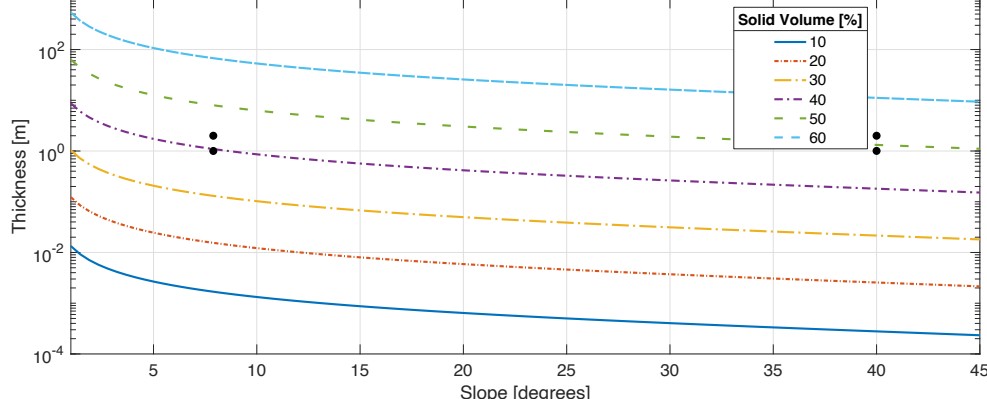


*Figure 3. Critical thickness as a function of topography slope and solid volume fraction computed with the following*
*values for the yield strength parameters: $a_2 = 0.272$; $b_2 = 22.0$. The four black dots represent couples of slope and*
*thickness values used to test the capability of the numerical solver to properly reproduce the triggering conditions of*
*lahars.*
**2.1.3 Erosion term**
Following the parameterization by Fagents and Baloga (2006), we adopted an empirical relationship for the volumetric
erosion rate $E_{tot}$ of the substrate:

$$E_{tot} = \epsilon h \sqrt{u^2 + v^2}(1 - \alpha_s). \tag{14}$$





This relation states that erosion is proportional to the thickness of the flow, the modulus of flow velocity and the
volumetric fraction of water in the flow, through an empirical constant $\epsilon$ (with units 1/[L]). In the original work by Fagents
and Baloga (2006), it is assumed that the rate of turbulent entrainment diminishes with increasing flow density. In fact,
as the flow entrains solid sediment, turbulence is progressively dampened (Costa, 1988). Here, because the density is
linearly proportional to the water volume fraction, we directly introduced a dependence of the erosion rate on this variable.
From the total erosion rate, we compute the entrainment rates of the solid phases, which are then used in the governing
equations, as:

$$E_{i_s} = \beta_{i_s}(1 - \alpha_d)E_{tot} \tag{15}$$

where $\beta_{i_s}$ are the relative volumetric fractions of the solid particles in the erodible substrate ($\sum \beta_{i_s} = 1$). When erosion
occurs, not only solid particles are entrained in the flow, but also the water present in the deposit, here assumed to saturate
its voids. This water entrainment from the erodible substrate is given by:

$$E_w = \alpha_d E_{tot}. \tag{16}$$


**2.1.4 Sedimentation term**

Sedimentation of particles from the flow is modelled as a volumetric flux at the flow bottom and it is assumed to occur
at a rate which is proportional to the volumetric fraction of particles in the flow and to the particle settling velocity $w_s$:

$$D_{s,i_s} = \alpha_{s,i_s} \cdot w_{s,i_s}(d_{s,i_s}, \rho_{s,i_s}, \nu_m). \tag{17}$$

The particle settling velocity $w_{s,i_s}$ is a function of the particle diameter $d_{s,i_s}$, the particle density $\rho_{s,i_s}$ and the mixture
kinematic viscosity $\nu_m = \frac{\mu_m}{\rho_m}$, and it is obtained by solving the following non-linear equation:

$$w_s^2(d_{s,i_s})C_D(Re) = \frac{4}{3}d_{s,i_s}g\left(\frac{\rho_{s,i_s} - \rho_a}{\rho_a}\right).$$

The gas-particle drag coefficient $C_D$ is a function of the particle Reynolds number ($Re = \frac{d_{s,i_s}w_s}{\nu_m}$), and it is calculated by
assuming spherical particles (although in the future can be generalized for more realistic shapes; Bagheri and Bonadonna,
2015; Dioguardi et al., 2016) through the following relations (Gidaspow, 1994):

$$\begin{cases} C_D = \dfrac{24}{Re}(1 + 0.15Re^{0.687}) & Re \leq 1000, \\[2mm] C_D = 0.44 & Re > 1000. \end{cases}$$


The dependence of the Reynolds number on the mixture kinematic viscosity acts on the settling velocity as a sort of
hindered settling. In fact, mixture viscosity increases with the total volumetric fraction of solids, and thus the settling
velocity decreases. This approach is described in Koo (2009), where several effective-medium models are analysed for
determining settling velocities of particles in a viscous fluid. Effective-medium theories have been developed for





predicting the transport properties of suspensions consisting of multiple particles in a fluid. In particular, the sedimentation
velocity is computed using the effective viscosity of the suspension, instead of the viscosity of the continuous phase.

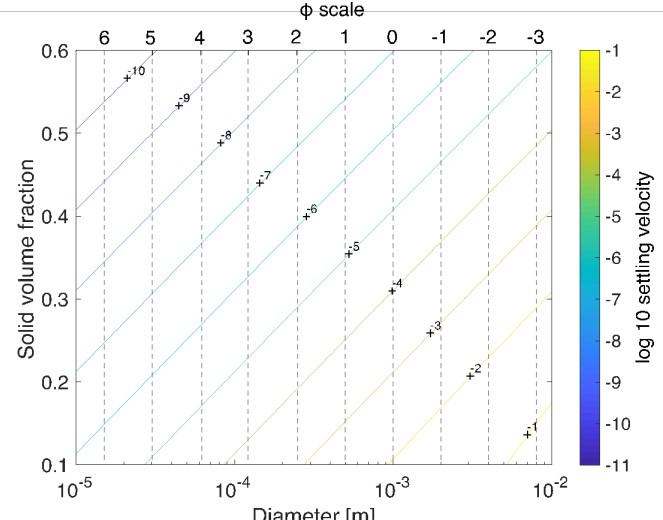


*Figure 4. Effective settling velocity. Values of the settling velocity are represented by the different contours, as a function*
*of particle diameter and total solid volume fraction.*
When considering the settling of solid particles, it is important to remind that we assume the flow deposit formed because
of sedimentation being saturated in water, with the volume fraction of water given by $\alpha_w$. Thus, the lahar does not loose
solid particles only because of sedimentation, but water too, with the volumetric deposition rate of water related to that
of solid particles by the following equation:

$$D_w = \frac{\alpha_d}{1 - \alpha_d} \sum_{i_s=1}^{n_s} D_{s,i_s}. \tag{18}$$


**2.2 Numerical implementation**



The numerical solution of the equations is based on the algorithm developed by de' Michieli Vitturi et al. (2019) for the
code IMEX-SfloW2D, and in particular on an operator splitting technique, where the advective, gravitational, and friction
terms governing the fluid-dynamics of the lahar are integrated in one step, while the erosion and deposition terms are
integrated in a second step. This allows ad-hoc numerical methods to be used for the different physical processes,
optimizing and simplifying the overall solution process.
The numerical integration of the advective, gravitational and frictions terms is based an Implicit-Explicit (IMEX) Runge-
Kutta scheme, where the conservative fluxes and the gravitational terms are treated explicitly, while the stiff terms of the
equations, represented by friction, are integrated implicitly. For the explicit spatial discretization of the fluxes, a modified
version of the finite-volume central-upwind Kurganov and Petrova (2007) scheme has been adopted. The scheme,
described in de' Michieli Vitturi et al. (2019) and Biagioli et al. (2022), has a second-order accuracy in space and
guarantees the positivity of the flow thickness. The spatial accuracy is obtained with a discontinuous piecewise bi-linear
reconstruction of the flow variables, in order to compute their values at the sides of each cell interface and thus the





numerical fluxes. The slopes of the linear reconstructions of flow variables in the *x*- and *y*-direction are constrained by
appropriate geometric limiters, allowing switching between low and high-resolution schemes.
The implicit part of the IMEX Runge-Kutta scheme is solved using a Newton-Raphson method with an optimum step
size control, where the Jacobian of the implicit terms is computed with a complex-step derivative approximation. The use
of an implicit discretization of the stiff friction terms allows for larger time steps, controlled by the CFL condition,
establishing a relationship between time step, flow velocity and cell sizes.
After each Runge-Kutta procedure, the erosion, deposition, and air entrainment term are integrated explicitly and the flow
variables and the topography at the centres of the computational cells are updated.
The numerical scheme is also designed to be well-balanced, i.e. to correctly preserve steady-states. This property is
important for the numerical simulation of lahars, for which the flow should be triggered only when the gravitational force
exceeds the frictional forces, and thus a proper balance of these terms must exist also in the discretized equations resulting
from the numerical schemes.

## 244    3 Model validation and applications


In this section we present a few applications of the proposed lahar model aimed at showing its robustness, applicability,
and performance. Concerning the numerical tests aimed at demonstrating the mathematical accuracy for the code
verification, the reader is addressed to de' Michieli Vitturi et al. (2019) where the code IMEX-SfloW2D, on which our
model is based, was presented. Applications of the code to hazard assessment for lahars in the Neapolitan area will be
presented in the companion paper by Sandri et al. (2023).
Firstly, we present the case of a lahar flow on a synthetic topography in order to investigate the triggering conditions.
Secondly, we introduce and describe all the needed variables to perform an application on real topography, that is the
Valle di Avella, one of the Apennine valleys adjacent to Mt. Vesuvius, where in the companion papers by Di Vito et al.
(this issue) and Sandri et al. (this issue) we also performed geological investigations and hazard analysis for lahar. In such
test area we explore the effects that can potentially affect the results, such as computational grid size, numerical scheme
order, water temperature, discretization of the grain size distribution, and erosion and deposition terms. As the two latter
processes are by far the most relevant on the key output variables such as run-distance, flow thickness and speed, in the
last subsection we use field observations to calibrate erosion and deposition terms.

## 260    3.1 Simulations on a synthetic topography: lahar trigger conditions


The first set of simulations we present is aimed at testing the capability of the numerical code to properly reproduce the
triggering conditions of a lahar, in terms of the relationship between initial thickness, solid fraction and slope. As
previously stated, the values of the friction parameters controlling the yield strength define a unique relationship between
thickness, slope and solid fraction resulting in a threshold for the mobility of the flow (see Fig. 2).
For the tests we consider a high- and low-angle slope (5 and 40 degrees respectively) and two values of the initial thickness
(1 and 2 meters) with different values of the solid fraction (30% and 40%).
The topography has a constant slope for *x*<0 m and is flat for *x*>0 m. In the left region of the domain, from x=-55 m to
x=-50m, the topography is excavated with a constant depth (1 or 2 meters). Then, from x=-50m, this region is connected
to the original topography with a quadratic function, in order to have a smooth transition and a horizontal slope at the
right end. The excavated volume is then filled with the liquid/solid mixture. In this way the free surface elevation of the



initial volume corresponds to the original topography elevation. The topography and the free surface are shown in the
panels of Fig. 5 with cyan and orange solid lines, respectively.
For this suite of tests, both erosion and sedimentation are neglected, in order to have a constant solid volume fraction
during the simulations and thus a better understanding of its effect on flow mobility. For all the simulations done, we
present in Fig. 5 the solutions in terms of the free surface of the flow at $t = 100$ s, corresponding to a steady state. In panel
(a) the final solution obtained for a slope of 7 degrees, an initial thickness of 1 m and a solid volume percentage of 40%
is shown. By looking at the diagram presented in Fig. 3, we can see that the black marker for this combination of slope
and thickness lies below the critical curve for 40% solid (purple line), thus the gravitational forces are smaller than the
yield strength and the initial volume should not move. Indeed, this is what happens in panel (a), even if a careful analysis
shows that on the left part of the volume there is a small change in the final free surface with respect to the initial constant
slope. This is an effect of the grid discretization, which results in a large slope for a very small area, over which the flow
is mobilized.
Panel (b) shows the final solution for the same condition as panel (a), except the initial thickness increased to 2 m. For
this thickness, and for a slope of 7 degrees, the marker in Fig. 3 is above the critical curve for 40% solid (purple line), and
thus the yield strength of the initial volume does not exceed the gravitational force. The liquid/solid mixture in this case
is mobilized with a small runout of a few meters at t=100 s. Both the flow thickness and the free surface slope decrease,
leading to a new steady condition reached when the flow momentum is dissipated by the friction forces.
Flow mobility increases also by decreasing the solid fraction. This is shown in panel (e), representing the final solution
for the same condition as panel (a), except for the solid volume percentage, lowered from 40% to 30%. By looking at the
diagram presented in Fig. 3, we can see that for this combination of slope and thickness the black marker lies well above
the critical curve for 30% solid volume (yellow line). In fact, the mixture moves along the slope and is able to reach the
topography break in slope, where most of the initial volume has reached a stable condition at $t = 100$ s. We observe that a
small portion of the flow is left at the base of the excavated area.
In the right panels of Fig. 5, a similar analysis is presented for a slope of 40 degrees. The first two simulations we present
are done with 50% solid volume (Fig. 3, green line), and initial thickness slightly below (1 m) and above (2 m) the critical
thickness for flow mobility. These initial conditions are represented by the right markers in Fig. 3. Fig. 5 (b) shows that,
as expected, with an initial thickness of 1 m the flow does not move and at $t = 100$ s the free surface has not changed with
respect to the initial condition, represented by the free surface parallel to the unmodified topography. When the initial
thickness is increased to 2 m (Fig. 5d), the flow starts to move with a final runout of a few meters only at $t = 100$ s, because
of the high yield strength associated with the large solid fraction. An initial thickness of 1 m, associated with a 30% solid
volume, results in a flow capable of moving along the 40 degrees slope leaving no deposit behind it, as shown in Fig. 5
panel (f). In fact, in this case, almost the whole initial volume reaches the flat part of the topography, with a long runout
and a thin deposit due to the speed gained by the flow on the high-slope region.







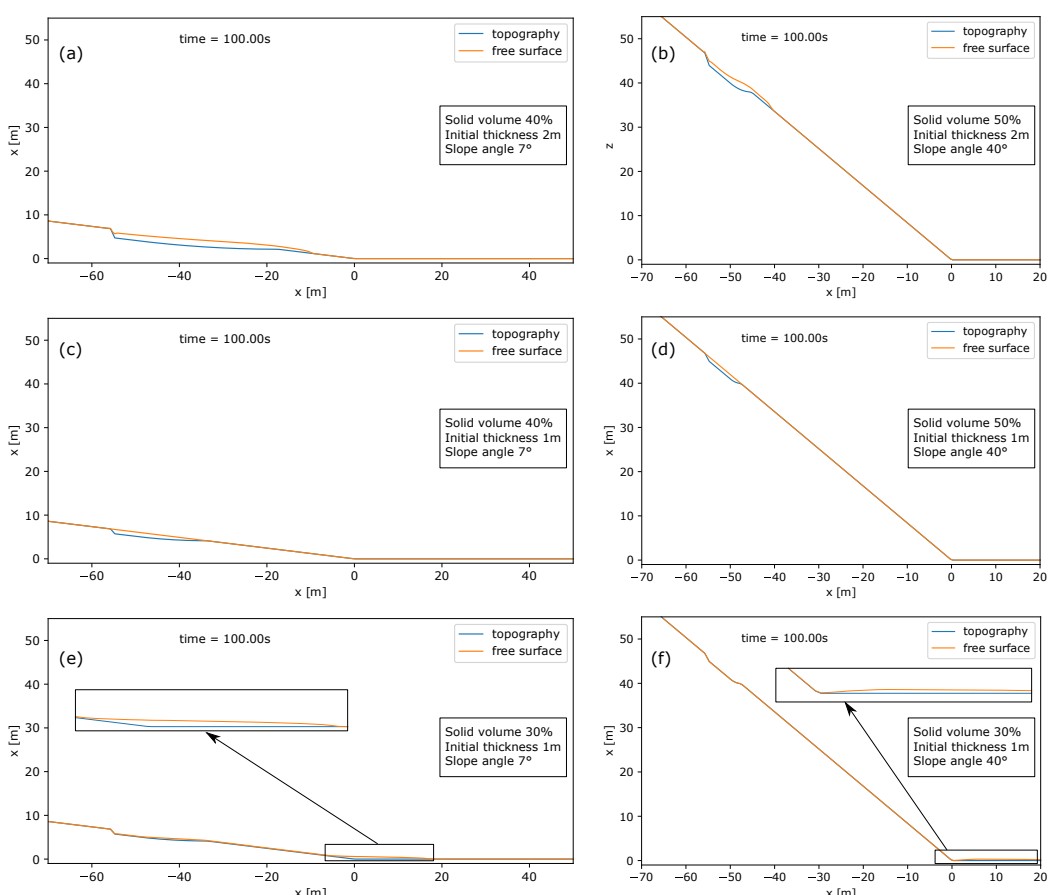


*Figure 5. Flow free surface (red line) and topography (blue line) for 6 simulations with different initial solid volume and*
*thickness and different slope.*

**3.2 Application to real topography: variables definition**

As an application of the model, we consider a syn-eruptive lahar from a medium-sized eruption at Somma-Vesuvius, that
is characterized by a total erupted mass between $10^{11}$ and $10^{12}$ kg (Macedonio et al., 2016; Sandri et al., 2016). To this
aim, as test case, we selected a synthetic deposit taken from one of the tephra fallout simulations made using the code
Hazmap (Macedonio et al., 2005) presented by Sandri et al. (2016); we considered a lahar generated by heavy rainfall
and we modelled the dynamics of the lahar in the Valle di Avella. In Sandri et al. (2016) a large number of tephra fallout
simulations were performed for a Probabilistic Volcanic Hazard Analysis by varying the wind field and the size and
intensity of the eruption. Among those, we selected a simulation that produced a substantial deposit (of the order of a few
decimetres) on the Apennine flanks facing the Valle di Avella. The Eruption Source Parameters associated to this
simulation are an eruptive-column height equal to 10.9 km, a mass eruption rate equal to $2.89 \times 10^6$ kg/s, a duration of the



fallout phase of 10 hours and total erupted mass as tephra fallout equal to $1.04\times10^{11}$ kg. The wind conditions are those
reported in the ERA5 reanalysis database for 14 December 2001.
For a correct modelling of the areas invaded by lahars it is necessary to use a digital terrain model (DEM) as accurate as
possible, such as that described in the companion paper by Sandri et al. (2023) which is used for this application.
For real life applications, a critical element in the definition of the initial conditions of a syn-eruptive lahar is the proper
identification of the areas of the topography where a lahar can be triggered, and the lahar's initial volume.
As regards the former, as already seen, the terrain slope is a key factor. On the basis of empirical observations, we assume
that lahars cannot be generated if the slope is: (i) less than a minimum threshold angle for remobilization ($\theta_{min}$), or (ii)
greater than an upper threshold angle ($\theta_{max}$), which prevents the accumulation, during the deposit phase of fallout
material, and which therefore cannot be remobilized by rainfall later to generate a lahar. The slope angle $\theta_{max}$ is fixed
here at 40 degrees (Bisson et al. 2014). As explained in the companion paper by Sandri et al. (2023), the value of the
lower threshold depends on the local granulometry and other factors that are necessary to be considered for a hazard
quantification in order to consider the uncertainty associated with this parameter. For this application we fixed $\theta_{min} =$
$30°$. Thus, on our computational grid we consider as possible source of the lahar only the cells with a slope between 30°
and 40°.
As regards the initial lahar volume, this is a consequence of the initial remobilization thickness $h_{tot}$ (see Figure 6 for a
graphical representation of the variables related to thicknesses and porosity) and of the area of remobilization. In turn,
$h_{tot}$ mostly depend on two parameters:

       ● the thickness of available compacted deposit, $h_s$ (i.e., devoid of the water filling its pore); in this application the


fallout deposit thickness is given by the ground tephra load provided by the Hazmap simulation, and selected

from Sandri et al. (2016);

● the amount of available water, denoted by $h_r$. Analysing the time series of rainfall at the OVO station located at

the historical site of the Vesuvian Observatory since 1940 and the data shown by Fiorillo and Wilson (2004), the

maximum rainfall was of the order of few tens of cm (the maximum recorded was 50 cm fallen in 48 hours near

Salerno on 26-10-1954). For this application we set the thickness of rainwater available to mobilize the water-

saturated deposit to $h_r = 0.5$ m, i.e., equal to the maximum recorded value. We stress that this is a conservative

choice, since lahars can also originate with less rainwater available, but in such cases their initial thickness (and

thus, volume) will be smaller. However, we also acknowledge that we are not accounting for the expected

increases in the maximum rainfall in a few hours, due to global warming, that are becoming more and more

frequent during the current decade (Esposito et al.,2018, Vallebona et al., 2015).


Let us call $h_w$ the thickness of the water layer that we could extract from the water-saturated deposit; then $h_w$ and $h_s$ can
be respectively expressed as a fraction of the thickness of the water-saturated deposit, $h_d$ , which has a porosity $\alpha_d$ , as:

$$h_w = h_d \alpha_d \tag{19}$$


and

$$h_s = h_d(1 - \alpha_d) \tag{20}$$









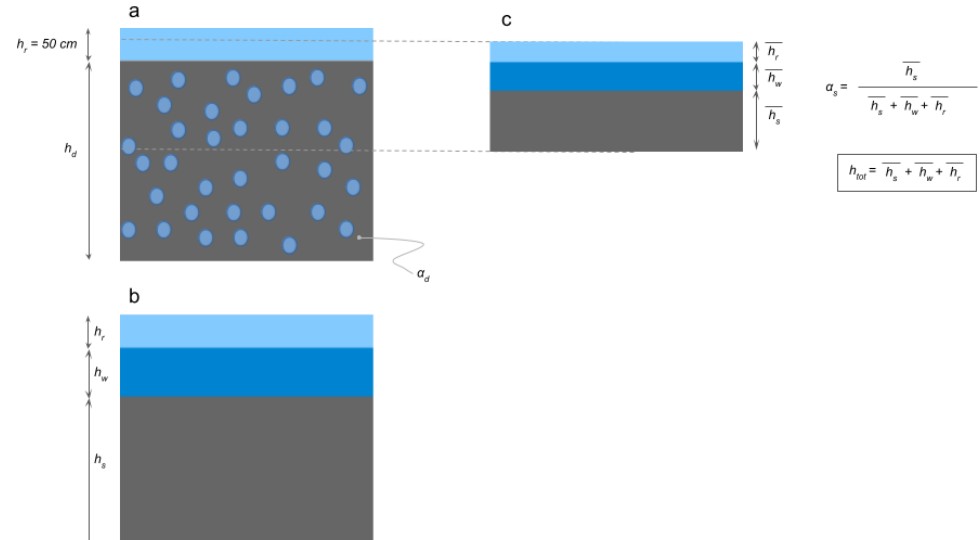

*Figure 6. Definition of the variables used to define the initial thickness mobilizable $h_{tot}$ . (a) The water-saturated deposit*
*of thickness $h_d$ , with porosity $\alpha_d$ , and the layer of rain water available of maximum thickness $h_r$ =50 cm (assumed). (b)*
*same as in (a) but if we imagine to extract all the pore-filling water and separate it into a layer of water of thickness $h_w$ ,*
*and a layer of compacted deposit of thickness $h_s$ , which is the tephra fallout deposit simulated by Hazmap simulator, in*
*this study. (c) the thickness of the mobilizable layers of deposit $\bar{h}_s$, rain water $\bar{h}_r$ and pore-filling water $\bar{h}_w$ depends on*
*the availability of rain and deposit and the fixed solid fraction αs of the initial flow.*


The initial flow thickness that is remobilized, $h_{tot}$ , will be the sum of three thicknesses:
• $\overline{h_s} \leq h_s$ from the solid part of the deposit,
• $\overline{h_w} \leq h_w$, from the water already filling the pores, and
• $\overline{h_r} \leq h_r$, from the rain (as said above, we assume $h_r = 0.5$ m).

There are relations linking these three addends. In particular, due to the condition of water saturation in the deposit

$$\alpha_d = \frac{\bar{h}_w}{\bar{h}_s + \bar{h}_w},$$

(21)

so that

$$\bar{h}_w = \frac{\alpha_d}{(1 - \alpha_d)} \bar{h}_s.$$

(22)

Moreover, in the initial flow volume there is a relationship between water and solid content, in terms of initial
volumetric fraction $\alpha_s$ :



$$\alpha_s = \frac{\bar{h}_s}{\bar{h}_s + \bar{h}_w + \bar{h}_r}, \tag{23}$$

so that (combining equation 22)

$$\bar{h}_r = \frac{(1 - \alpha_s)}{\alpha_s} \bar{h}_s - \frac{\alpha_d}{(1 - \alpha_d)} \bar{h}_s = \frac{1 - \alpha_d - \alpha_s}{\alpha_s (1 - \alpha_d)} \bar{h}_s \tag{24}$$

We see from equations 22 and 24 that both $\bar{h}_w$ and $\bar{h}_r$ are linear functions of $\bar{h}_s$. Considering the initial availability of
remobilizable deposits, we can state that

$$\bar{h}_w + \bar{h}_s \leq h_d \tag{25}$$

or, using equation 22,

$$\bar{h}_s \leq (1 - \alpha_d) h_d \tag{26}$$

Considering, on the other hand, the available water from rain, we have

$$\bar{h}_r \leq h_r \tag{27}$$

or, using equation 24,

$$\bar{h}_s \leq \frac{(1 - \alpha_d)\alpha_s}{(1 - \alpha_d) - \alpha_s} h_r \tag{28}$$

The maximum solid thickness $\bar{h}_s$ that can be remobilized, considering the availability of water-saturated deposits and
rain, and the a priori sampled initial solid fraction $\alpha_s$, is then the maximum satisfying both conditions in equations 26
and 28, i.e.:

$$\bar{h}_s \leq \min \left\{ \frac{(1 - \alpha_d)\alpha_s}{1 - \alpha_d - \alpha_s} h_r ; (1 - \alpha_d) h_d \right\} \tag{29}$$


Once this is known, we can get the total initial thickness of the lahar, by simply computing it as

$$h_{tot} = \frac{\bar{h}_s}{\alpha_s} \tag{30}$$


The ashfall deposit which does not contribute to the initial volume of the lahar is added to the pre-existing topography as
an erodible layer. The contribution of the ash fall deposits in the intermediate and distal areas has been significant in the
past sub-plinian eruptions, as shown in the paper by Di Vito et al. (this issue).
The steps described above are represented in Figure 7 for the real-topography test application to Valle di Avella, from the
identification of areas "prone" to remobilization on the basis of geomorphological features, the terrain slope (top panel,





red pixels), to the application of the criterion in equation 29 and 30 to compute the initial thickness of lahar (bottom panel)
from the rainwater available and the ashfall deposit (top panel, contour lines). For the case presented in Figure 7 we
assumed a deposit porosity $\alpha_d = 0.22$ and an initial solid fraction in the lahar $\alpha_s = 0.29$. With these values, equations
29 and 30 give, for an ashfall deposit thickness of 0.4m and an amount of rain of 0.5m, an initial lahar thickness of
approximately 0.8m.
Concerning the grain size distribution of the remobilized deposits here we used that obtained by Di Vito et al. (this issue)
on the basis of field data analysis.

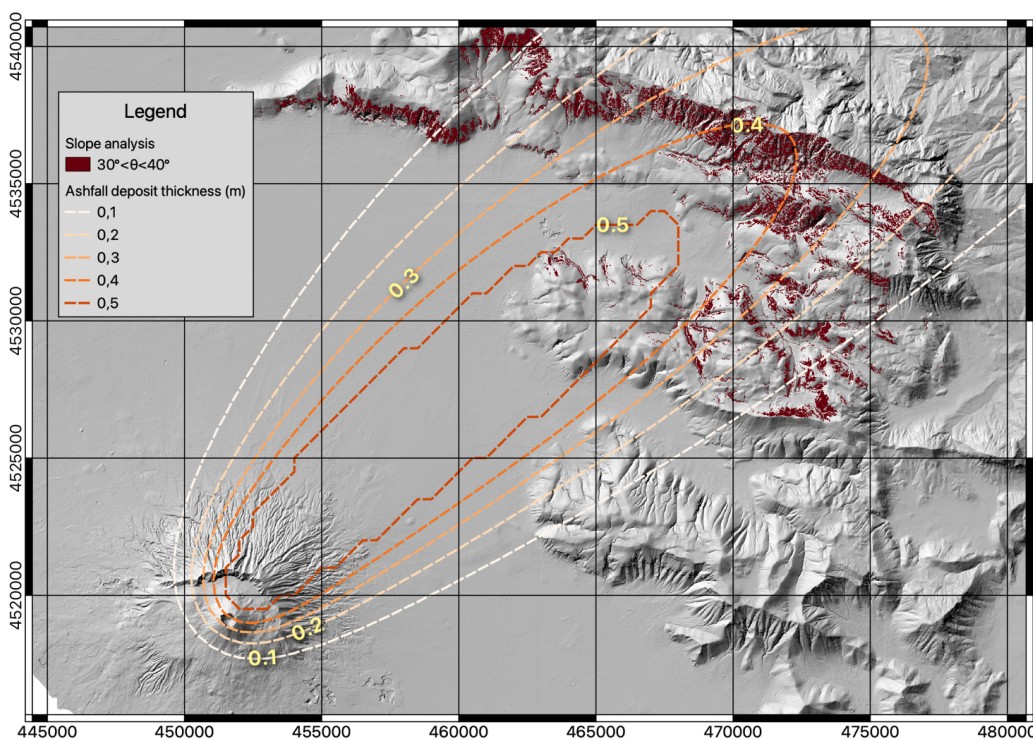



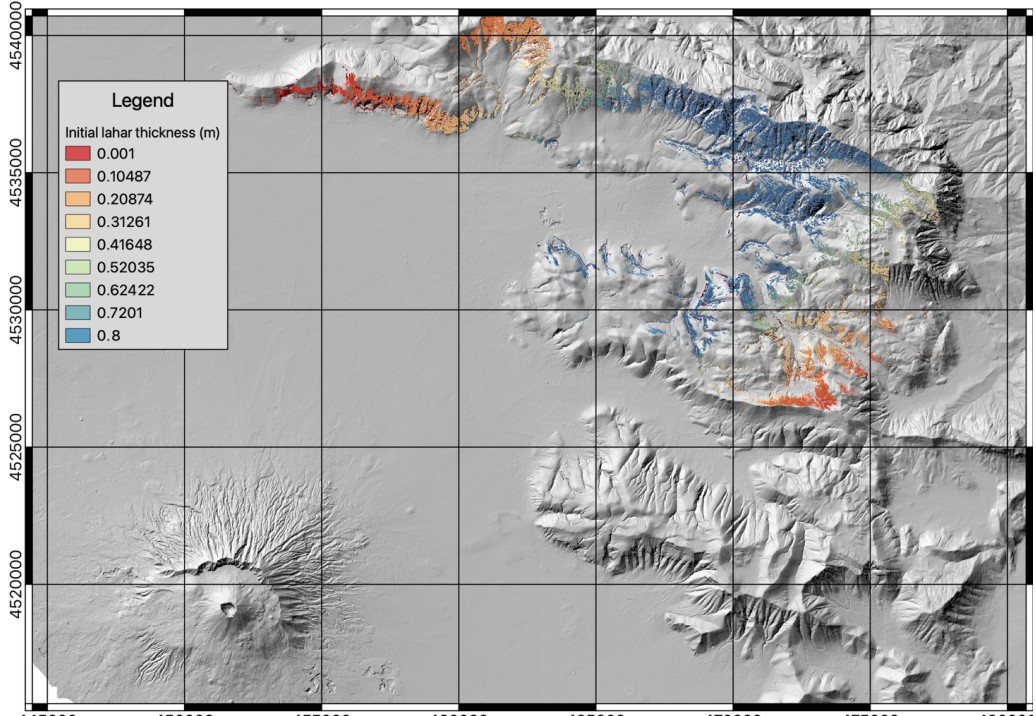


*Figure 7. Steps for the definition of the initial lahar thickness. The top panel shows grid cells with slope between $\theta_{min}$*

*and $\theta_{max}$ (red pixels) and the HAZMAP deposit thickness (contour lines). The bottom panel shows the initial lahar*

*thickness.*

**3.3 Application to real topography: sensitivity tests and description of the relevant output variables**

We conduct a series of sensitivity tests on the real-topography test area, in order to quantify the relevance of different
terms and processes on the output of the simulations, in terms of flow thickness and/or area.

We first present a reference simulation, extracted from the ensemble of simulations presented in Sandri et al. (this issue),
and for this case we show the temporal evolution of the flow and the most relevant output produced by the model. Then,
with respect to this simulation, we vary several parameters to show the sensitivity of the results to several model
parameters.

**3.3.1 Flow evolution and relevant output**

In this section we describe a reference simulation, obtained for a computational grid with cells of 50m and a second-order
numerical scheme in space, by applying a van Leer slope limiter to the reconstruction of the flow variable. For this
simulation, the total grain size distribution is discretized with 6 bins, from $\phi = -3$ to $\phi = -7$, and we assume an initial
temperature of the lahar of 373K. While we recognize that this temperature is more adequate for syn-eruptive lahars from
pyroclastic density current deposits, here we used this value to better show the effect of the temperature on the lahar
dynamics. In fact, later in the paper, we compare the results with those obtained with a colder lahar (300K).




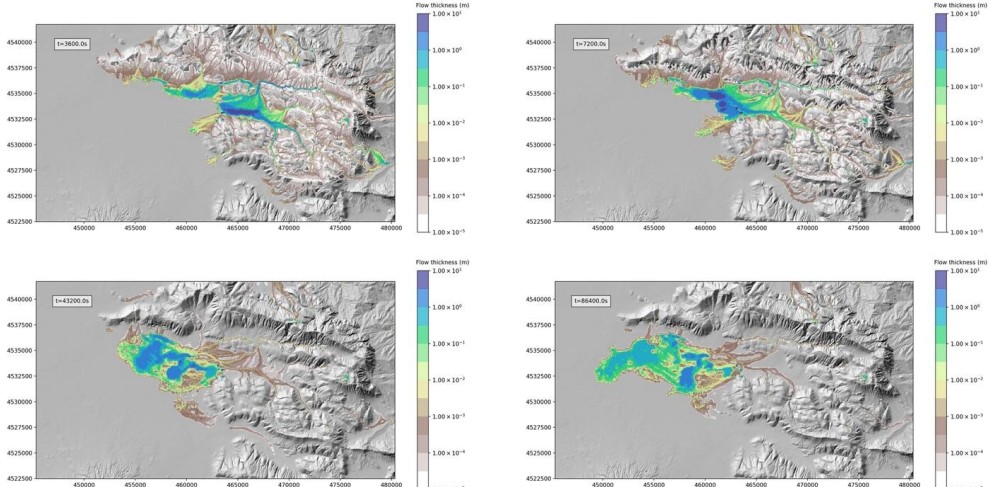

*Figure 8. Lahar thickness temporal evolution: top-left at 3600s; top-right at 7200s; bottom-left at 43200s; bottom-right*
*at 86400s.*
The initial thickness of the lahar is shown in the bottom panel of Figure 7, and its temporal evolution is presented in the
four panels of Figure 8. After one hour from the mobilization (Figure 7, top-left panel) the lahar already invaded a large
portion of the Valle di Avella, with its maximum thickness reaching a few meters in its southern part, and a thickness of
few millimetres still moving on the flanks of the Apennines facing the valley.
After 1 hour of flow time, the lahar has already reached the localities of Avella, Roccarainola and Camposano, which all
are inside the case-study valley, while after 2 hours the lahar has reached the city of Nola, just outside the valley. After
12 hours of flow time, the lahar has already reached the localities of Marigliano and Cancello Scalo, the first being in the
more open plain, while the second nearby the NWW Apennine sector of the valley. After 24 hours of flow time, the lahar
has already reached the city of Acerra in the open plain. Although this simulation is not aimed at reproducing a particular
event from the past, but at showing the model's ability to describe the different phenomena that may characterize a future
lahar in the Avella Valley, it is interesting to note that these reaches are corroborated by some historical sources on the
"1631" events, for which it is reported that the localities of Marigliano and Nola were reached by those lahars and by
geological evidences reported in Di Vito et al. (this issue).



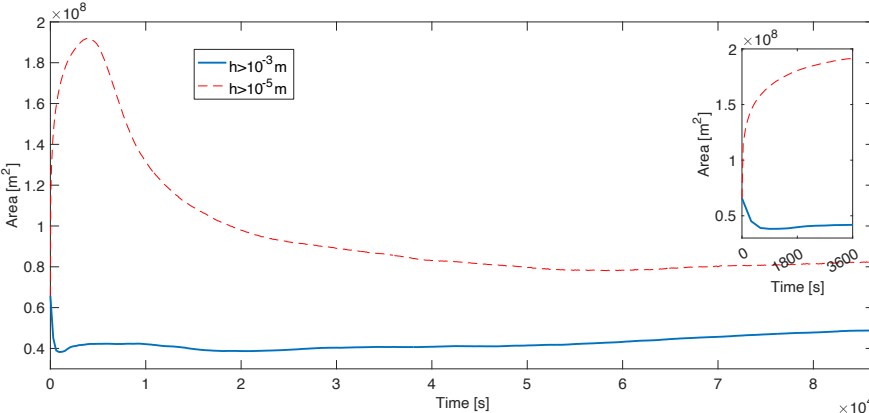

*Figure 9. Area of the lahar versus time for the reference simulation. For the computation of the area two thresholds on thickness have been applied: a physical one (solid blue line, $h \geq 10^{-3}$ m), and a numerical one (dashed red line, $h \geq 10^{-5}$ m). The inset represents a detail of the first hour of simulation.*

The area invaded by the lahar changes with time and its evolution is presented in Figure 9. The model computes at each time step the invaded area as the sum of the areas of the grid cells where flow thickness is greater than a fixed threshold. For this analysis, two thresholds on the minimum flow thickness have been applied, a "physical" threshold set to $10^{-3}$m (represented in Figure 8 by the solid blue line), which allows to analyse the dynamics of the bulk of the lahar, and a "numerical" threshold set to $10^{-5}$m (represented by the dashed red line). It is important to remark that such a small threshold does not correspond to a thickness for which the flow is properly described by our model equations, because for such values forces like surface tension becomes larger than gravity and friction (Hong at al., 2016). In any case, this small threshold can provide information on the dynamics of the very thin tail of the lahar, where the velocity goes rapidly to zero because of friction forces. Figure 9 shows that, for the larger physical threshold, at the beginning of the simulation (first 15 minutes) there is a rapid decrease in the area, due the channelization phase of the flow mobilized from the flanks of the Apennines. After this initial phase, the flow reaches the Valle di Avella and starts to spread out, with the area of the lahar increasing with time. For the lower thickness threshold, we observe that the area rapidly increases during the initial slumping phase of the lahar and it reaches its maximum after approximately 1 hour after the mobilization. Then it decreases, first rapidly and then more slowly, increasing again after 15 hours. This is due to the fact that tail of the flow gets thinner with time and, as previously described, the presence of the yield strength term in the friction allows the flow to stop with a thickness that depends on the slope of the topography and on the fraction of solid material left in the flow. Thus, when the thickness is small enough, the tail of the lahar slows down and stops moving. Because of that, erosion becomes negligible and at the same time deposition occurs, further increasing the thinning of the deposit and the loss of sediments and water by the flow. This is well shown by the evolution of flow thickness on the flanks of the Apennines, as illustrated in Figure 8. After one hour from the mobilization, thickness is less than 1 millimetre and, for the slope of the Apennines and the water content of the flow, this value is well below the critical thickness (see Figure 3). Because of that, the flow stops to move and the only process occurring is the loss of water and sediments.



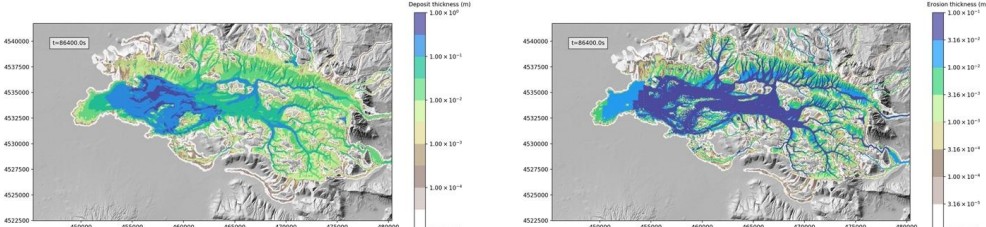

*Figure 10. Total deposition (left) and erosion (right) after 24 hours of simulation.*
The mobility of the flow is mostly controlled by the solid fraction within the lahar, and this fraction can change because
of erosion and deposition. Thus, the total erosion and deposition are important factors controlling the area invaded by the
lahar. The final deposit and erosion thickness are presented in the left and right panels of Figure 10, respectively, showing
a significant erosion where the flow is channelized, reaching a maximum value of a few decimetres. Conversely,
deposition mostly occurs in the flat areas invaded by the lahar where the flow slows down, producing a maximum deposit
thickness of the order of 1m.

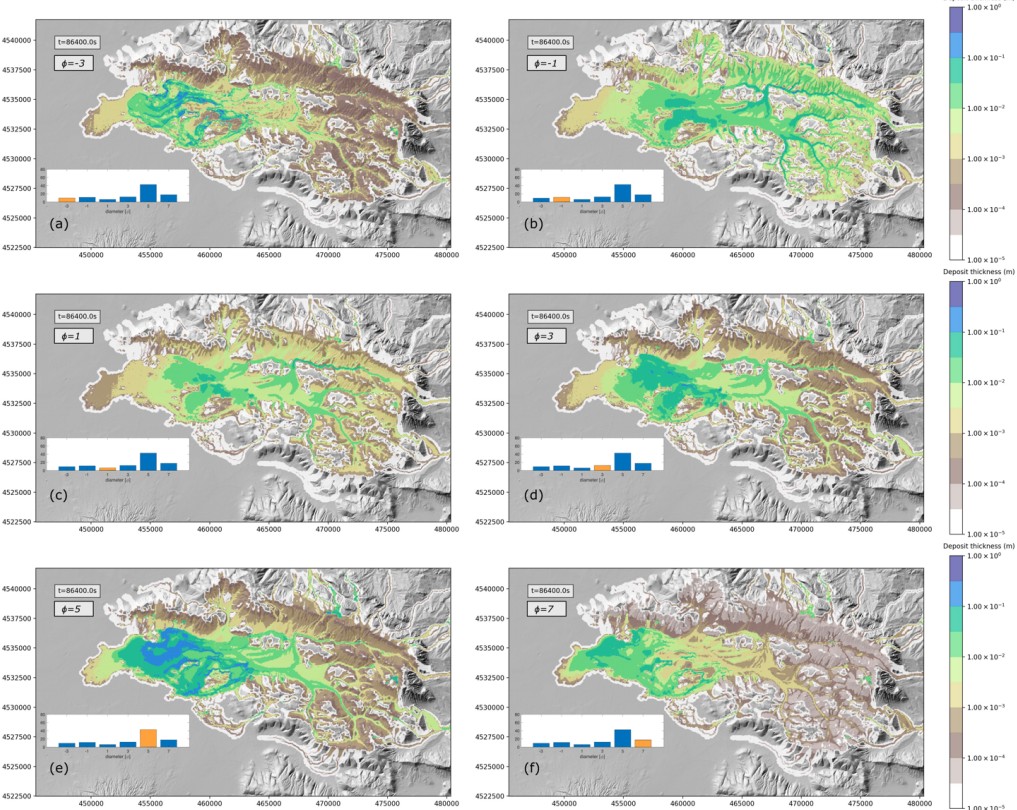


*Figure 11. Total deposit thickness after 24 hours of simulation for the 6 different classes of particles: (a) $\phi = -3$; (b)*
*$\phi = -1$; (c) $\phi = 1$; (d) $\phi = 3$; (e) $\phi = 5$; (f) $\phi = 7$. The insets in each panel show the initial total grain size*
*distribution of the lahar, and the class for which the deposit is shown in the panel is represented in orange.*






As shown by equation 17, deposition is proportional to the settling velocity of the sediments, which increases with their
sizes. This reflects in different depositional patterns for the different classes of particles, shown in the panels of Figure
11. We observe that the thickness of the deposit for the different classes does not depends only on the settling velocities,
but also on the amount of sediments available for deposition, and thus on the initial grain size distribution of the lahar.
This explains why the larger contribution to the deposit is given by class $\phi = 5$, for which the maximum thickness deposit
24 hours after the mobilization of the lahar is about 1 m. For classes $\phi = -3$ and $\phi = -1$ the initial mass fractions are
similar, and the difference in the final deposit is mostly due to the differences in settling velocities. In fact, Figure 4 shows
that, for the same total solid volume fraction of the lahar, a difference in size in the Krumbein scale of $2\phi$ results in a
difference in the settling velocity, and thus in the deposition rate, of one order of magnitude.

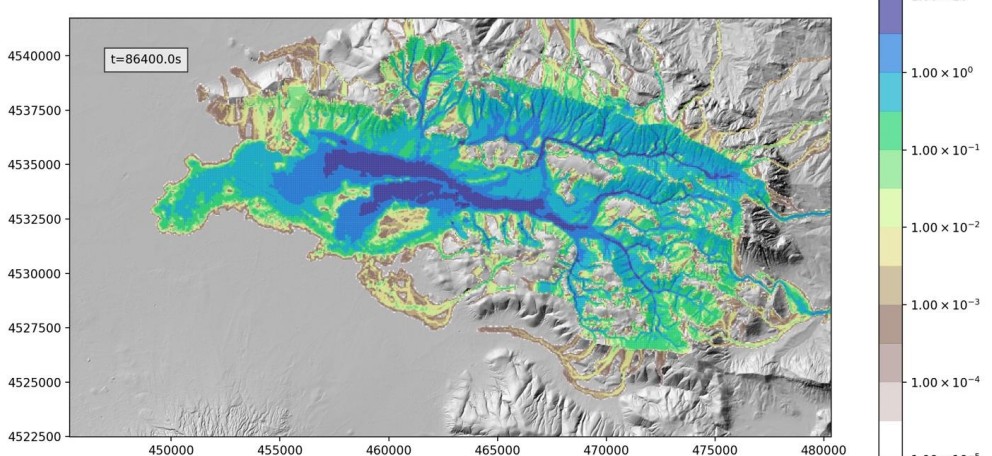


*Figure 12. Maximum thickness of the flow in each cell of the computational grid during the 24 hours of simulation.*
From the perspective of hazard assessment, it is not important the flow thickness at the end of the simulation (here 24
hours after the mobilization), but the maximum thickness registered at each location reached by the lahar in the same time
span, as shown in figure 12. This figure shows that the maximum thickness can exceed several meters over a large area
of the domain, allowing to identify the areas where the hazard is significant.

**3.3.2 Effects of grid size and numerical scheme order**
In this section we want to present the effects of the resolution of the computational grid and of the spatial numerical
scheme adopted (first and second order schemes). We remind that the DEM resolution used for the simulations is 10 m,
while the computational grid resolution used for the reference simulation presented in the previous section was 50m.
Thus, the smaller topographical features present in the original DEM are smoothed in the computational grid, possibly
with an effect on the dynamics of the simulated flow. Here, we focus our interest to the first 2 hours of the simulation,
thus the phase where the details of the topography can be more important, because of the important canalization effects
acting on the lahar when moving down the flanks of the Apennines into the Valle di Avella. All the simulations for this
analysis have been performed on 16 cores of a Multicore shared memory server SuperMicro 4×16-core AMD 2.3 GHz.

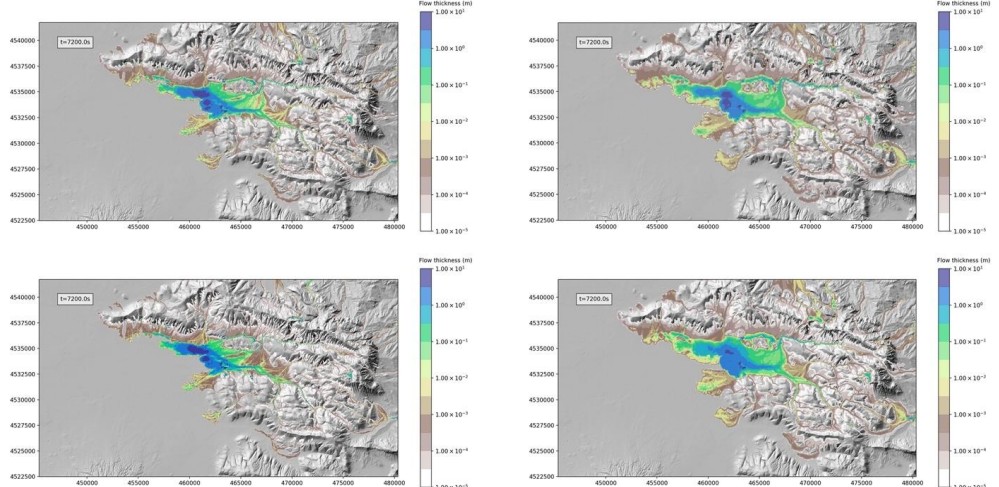

*Figure 13. Maps of flow thickness at t=7200s for simulations with different grids or different numerical schemes: (a)*
*50m grid resolution and 2nd order scheme with geometric limiter; (b) 100m grid resolution and 2nd order scheme with*
*geometric limiter; (c) 25m grid resolution and 2nd order scheme with geometric limiter; (d) 50m grid resolution and*
*1st order scheme (no geometric limits used).*

In Figure 13 we compare the flow thickness of the reference simulation (top-left) with: a simulation obtained with a 100m
resolution computational grid (top-right); a simulation obtained with a 25m resolution computational grid (bottom-left);
a simulation with a 50m resolution computational grid but with a first-order spatial scheme (bottom-right). While there is
a remarkable difference in the area invaded by the flow between the reference 50m simulation and the 100m simulation,
the difference between the reference simulation and the 25m one, in particular for significant flow thicknesses, is very
small. We also have to account that theoretically the computational time required for a simulation, when the grid cell size
is decreased by a factor 2, increases by a factor $2^3$. In fact, the number of horizontal cells increases by a factor $2^2$, being
the simulation two-dimensional, and the time step decreases by a factor 2, due to the well-known linear relationship
between spatial and temporal step associated with the use of an explicit integration scheme (CFL condition, Courant et
al. 1928). In addition to this, the CPU time required for the initialization of the arrays and for the input/output procedures
must be accounted. For this particular case, the 100m, 50m and 25m resolution simulations required 1023s, 6916s, and
50289s, respectively. This suggests that, with the DEM we used, a 50m resolution is adequate for a proper description of
the flow dynamics, also in view of the utilization of the simulations for hazard studies, where a large number of runs is
required and the computational time is an important constrain.
Finally, in the bottom-right panel of Figure 13, we can see the output of a simulation with the same resolution of the
reference one (50 m), but without the use of geometric limiters for the linear reconstruction of flow variables at the
interfaces of the computational cells. This makes the discretization scheme of first order, with respect to the second order
obtained for the reference simulation. The difference in the results is striking, with the first order simulation being more
similar to the simulation obtained with the 100m grid, and the second order simulation being similar to that obtained with
the 25m grid. The computational overhead associated with the use of geometrical limiters is small (6916 seconds vs 6770
seconds), thus their use is strongly suggested for this kind of simulations.



### 3.3.3 Effects of grain size discretization

In this section we present the sensitivity of model results to the discretization of grain size distribution. With respect to the reference simulation, where 6 classes were used, here we compare the solution after 4 hours from the mobilization of the lahar with those at the same time for two simulation with the total grain size distribution described by 3 and 12 particle size classes, respectively. The results of this analysis are presented in figure 14, with the final flow thickness presented on the left panels and the deposit thickness on the right panels. The plots show small differences between the simulations with 3 (figure 14 a-b) and 6 classes (figure 14 c-d), which become almost negligible when comparing the simulations with 6 and 12 classes (figure 14 e-f). For this test case, the increase in the number of classes, from 6 to 12, resulted in an increase of the computational time of a factor 1.3. Thus, the choice of using 6 classes for the reference simulations represents a good compromise between accuracy and efficiency.

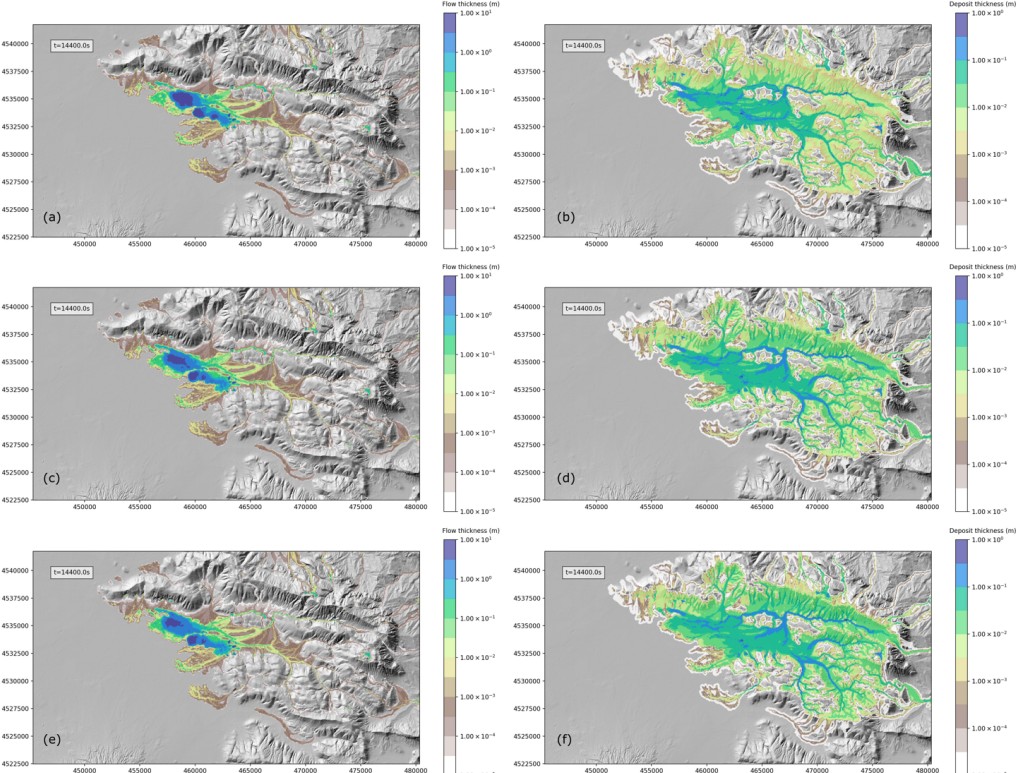

*Figure 14. Maps of flow thickness (left) and deposit thickness (right) at t=14400s for simulations with different discretization of the total grain size distribution: (a-b) 3 classes; (c-d) 6 classes; (e-f) 12 classes.*

### 3.3.4 Effect of initial temperature

In this section we present a comparison between the output of the reference simulation (T=373K) and a simulation with a lower initial temperature (T=300K).




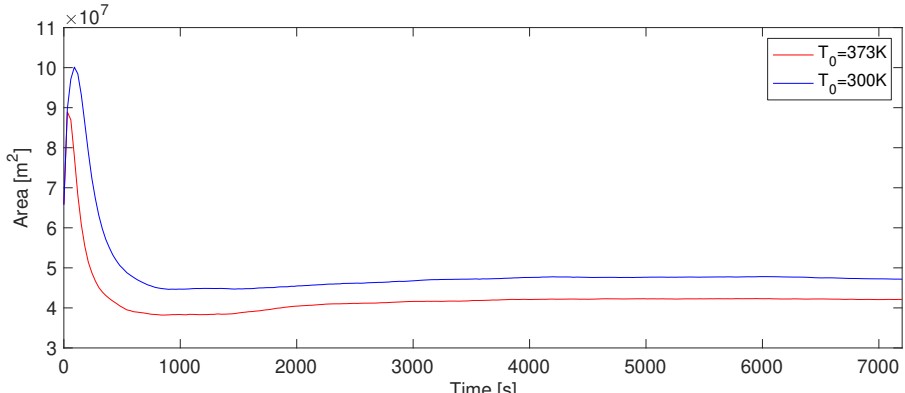

568

*Figure 15. Area of the lahar versus time for the simulations with different initial temperatures: 300K (blue line) and 373K*

*(red line). The area is computed as the sum of the areas of the grid cells where flow thickness is greater than $10^{-3}$m.*

Figure 15 shows the invaded area (computed the area as the sum of the areas of the grid cells where flow thickness is greater than or equal $10^{-3}$m) versus time for the two simulations, where the result for the reference simulation is presented with a red line, while the result for the colder case is plotted with a blue line. We remark that here we are not plotting the area of the deposit of the lahar, but the area where the lahar is still moving, in order to better understand how flow viscosity affects the dynamics of the flow. In the initial phase (<60s), the difference between the two cases is negligible, while it becomes more significant with time, with the area of the colder flow exceeding that of the reference one. This can seem counterintuitive, because we expect an increased mobility for the hotter flow due to the lower viscosity, and thus a larger runout. But the initial phase is dominated by flow channelization, which is increased by the larger mobility, and which results in a smaller footprint of the lahar. The different viscosity of the flow also affects the tail of the flow in a twofold way. Indeed, the lower viscosity results in a larger settling the velocity of the sediments and a debulking which further increases the flow mobility. This is evident by looking at the reduced footprint of the flow left on the Apennines flanks in the simulation with the higher initial temperature (Figure 12a).

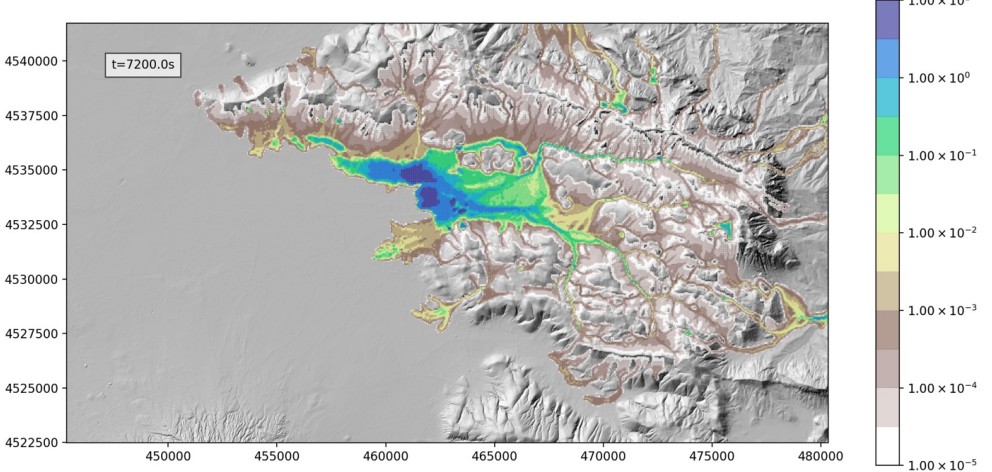


*Figure 16. Maps of flow thickness at t=7200s for a simulation with an initial temperature T=300K.*

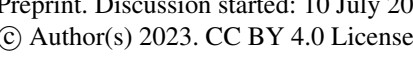




### 3.3.5 Effects of erosion and deposition

As shown in the previous comparison, viscosity of the flow has an effect on the debulking process, which in turn can
affect the lahar propagation. Here we focus our attention on the effects of the main processes controlling lahar bulking
and debulking, i.e. the deposition and erosion processes.
This is done by comparing in Figure 17 the first 2 hours of the reference simulation with 3 additional test cases: a
simulation without erosion (top-right); a simulation without deposition (bottom-left); a simulation without erosion and
deposition (bottom-right).
By comparing the flow thickness and the area covered by the flow of the reference simulation and that without erosion,
we can see the twofold effect of the bulking associated with erosion. On one hand we observe the larger flow thickness;
on the other hand, we observe a smaller runout, due to the lower mobility associated with a higher solid volume fraction.
This is particularly true in the Valle di Avella, where the front of the flow advanced about 2km more for the simulation
without erosion.

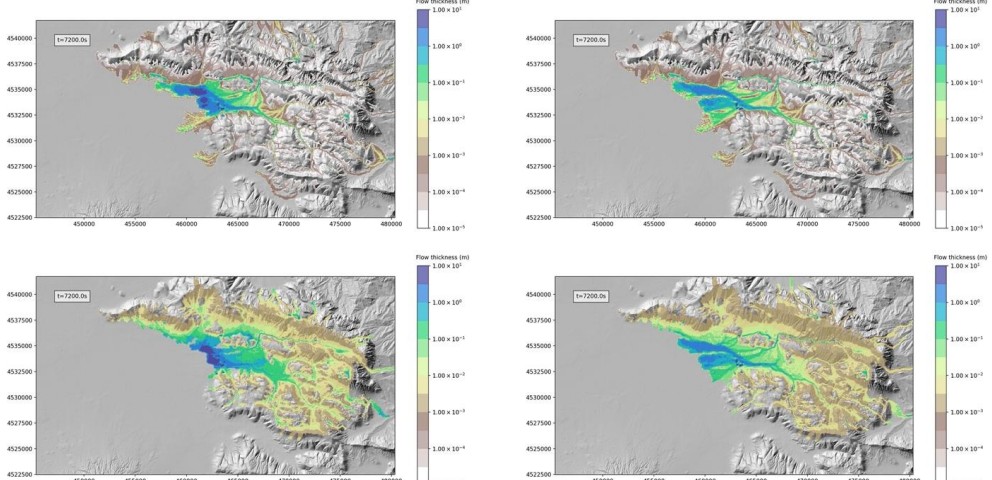

*Figure 17. Maps of flow thickness at t=7200s for simulations with and without erosion and deposition: (a) reference*
*simulation with erosion and deposition; (b) simulation with deposition and without erosion; (c) simulation with erosion*
*and without deposition; (d) simulation without erosion and deposition.*

### 4 Conclusions

A new shallow layer model for describing lahar transport was presented. The proposed model does not describe all the
general aspects of lahar behaviour (see Pudasaini, 2012) but contains the essential physics needed to reproduce the general
features of lahars observed in nature, crucial for assessing their hazard.
In particular the model considers realistic particle size distribution, surface erosion and deposition processes through
semi-empirical parameterizations calibrated from field data.
The model was developed with the aim to describe lahar propagation and deposits and assess their hazard in contexts
similar to that of the Vesuvius area, which is highly populated and prone to this kind of phenomenon after heavy rains
(e.g., Fiorillo and Wilson, 2004).



The critical variables were identified and several sensitivity tests carried out, using synthetic and real cases topographies.
The variables used in order to define the source are the initial mobilizable thickness, the water-saturated deposit thickness,
the layer of rain water, and the thickness of compacted deposit, which is related to the others through the substrate
porosity.
The steps used for the assessment of the initial lahar thickness were presented for the real-topography test application to
Valle di Avella.
The comparison of simulations obtained for different numerical grids (from 25 m to 100 m), scheme order, and grain size
discretization were useful to find a good compromise between resolution and computational speed. The used DEM was
however at a resolution (10 m) finer than that of the computational grid.
The friction term is defined as the sum of a velocity-independent yield slope, a viscous slope, and turbulent slope (O'Brien
et al., 1993). The yield strength and the fluid viscosity are considered functions of the total solid volumetric fraction in a
consistent way. The values of the three terms strongly depends on volumetric solid fraction, on flow thickness, and
velocity. They can vary in a non-linear way by several orders of magnitude when thickness, velocity and solid fraction
vary in ranges typical for lahars. This can produce a stiff term in the system of equations, and, for this reason, it is needed
a robust solver that allows the coupling between the gravitational and frictional terms to be accurately simulated.
Energy transport and temperature effects were also explored in order to better understand how flow viscosity affects the
dynamics of the flow. When the friction is dominated by the yield slope term, the difference between the high and low
temperature cases is negligible, while it becomes more significant with time, with the area of the colder flow exceeding
that of the cold one. In fact, the lower viscosity in the case of the hot flow, beside an increased mobility, results also in a
larger settling the velocity of the sediments and a debulking which further increases the flow mobility, producing a
reduced footprint deposit area of the flow.
Effects of erosion and deposition were investigated by comparing a simulations i) without erosion, ii) without deposition,
iii) without erosion and deposition, and iv) with erosion and deposition. By comparing flow thickness and area covered
by the flow, we can see the twofold effect of the bulking associated with erosion, that consists in larger flow thicknesses
and smaller runouts, due to the lower mobility associated with higher solid volume fractions.
The companion paper by Sandri et al. (this issue) will show an application of the presented model for hazard analysis of
lahars from Vesuvius deposits in the Neapolitan area.

**Author Contribution**
MdMV, AC and LS defined the set of governing equations of the model. MdMV, LS, AC, MDV and DD defined the
equations for the initial conditions. MdMV developed the code. MdMV, LS, AC, MDV and DD defined the set of
simulations and MdMV performed them. MdMV prepared the manuscript with contributions from all co-authors.

**Competing interests**
The authors declare that they have no conflict of interest.

**Acknowledgments**
This work has been produced within the 2012–2021 agreement between Istituto Nazionale di Geofisica e Vulcanologia
(INGV) and the Italian Presidenza del Consiglio dei Ministri, Dipartimento della Protezione Civile (DPC), Convenzione
B2. We also thank Marina Bisson and Roberto Gianardi for providing the DEM used for the simulations.



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
