# Peer review of "Lahar events in the last 2,000 years from Vesuvius eruptions. Part"

_EGUsphere, 2023_

## Referee Comment (RC1)

**Introduction**

In the present study, novel insights are presented that facilitate a comparative analysis of erosion and deposition processes in lahars evolution. Actually, flow thickness has been employed as a determinant factor. It might be instructive—though not imperative—to scrutinize the influence on dynamic pressure, particularly if a notable alteration is discerned. The manuscript is methodically robust and coherently structured. Both the objectives and the conclusions drawn are lucid and of significant import.

**Specific points**

- The abstract cogently delineates the content of the manuscript. Additionally, it references subsequent studies that both corroborate and expand upon the subject matter.
- The conclusions drawn are both transparent and congruent with the empirical data elucidated within the text. The assumptions and constraints of the model are appropriately outlined, thereby enabling the derivation of specific inferences concerning the subject at hand.
- Given that the code is accessible to the reader, the reproducibility of the obtained results is verifiable.
- The title aptly encapsulates the essence of the manuscript's content.
- The prose is articulate, and the utilization of equations and figures is both pertinent and accurate.
- Notwithstanding these strengths, certain technical modifications are requisite prior to the paper's publication.

**Technical comments**

1. Missing reference (citation in bibliography)
   a. Line 44 (IEA et al, 2016)
   b. Line 228 (Biagioli et al 2022) but bibliography says 2021
   c. Line 250 Is Sandri et al 2023 the same that Sandri et al (this issue)? If yes please change  Sandri et al 2023 citation, if not reference is missing
   d. Line 334 missing reference (Bisson et al 2014)
   e. Line 466 review reference (Hong at al, 2016) by (Hong et al.,2016)
2. Figures legend
   a. Figure 3 wrong legend (solid fraction title does not correspond with the caption and text)
   b. Please review Figure 5, the descriptions in the text  (from line 277 to 304) do not correspond to the assigned label (a,b,c,d,e,f) figures that seem to be switched.
   c. Lines 443 are talking about Figure 8 but Figure 7 is referenced
   d. Line 447 ¿Is it possible to show in Figure 7 the locations mentioned in line 447 ?
   e. Missing (a,b,c,d) in Figure 13
   f. Line 582 mention Figure 12a (does not exist)
   g. Missing (a,b,c,d) in Figure 17

---

## Author Comment (AC1)

Dear Editor,

We are writing to express our gratitude for the insightful and constructive comments provided by the reviewers for our manuscript titled "Lahar events in the last 2,000 years from Vesuvius eruptions. Part 2: Formulation and validation of a computational model based on a shallow layer approach" (Preprint egusphere-2023-1301). We are pleased to inform you that we have carefully considered the feedback and suggestions provided by Reviewer 1 and Reviewer 2 and have revised our paper in accordance with their recommendations.

Reviewer 1's comments highlighting the need to show the influence of dynamic pressure. In response, we discussed this point and added a new figure showing two dynamic pressure maps. Additionally, Reviewer 1 suggested several technical corrections which have been addressed.

Reviewer 2's feedback was mostly focused on the capability of the model to perform a counterfactual analysis of the 1631 event. As per this suggestion, we now discuss this point in the Conclusion section, with a reference to the work done in the companion paper Sandri et al. where we considered alternative hydraulic catchment, together with the variability in the initial volume, in the initial water fraction and initial thickness of the ash deposit. These addition helps to better put our paper in the context of risk analysis.

In addition to the suggested revisions, we have made some minor changes to the text, ranging from improvements in clarity and coherence to the correction of typographical errors. We also replaced the original figures with new ones with a higher resolution.

We believe that the revisions made in response to the reviewers' feedback have substantially enhanced the quality and impact of our manuscript. We are confident that the changes we have implemented align well with the standards of rigor and clarity that Solid Earths upholds.

Enclosed herewith, please find the detailed answers to all reviewer's comments, the list of new references added to the manuscript, and a revised version of our manuscript highlighting all the changes. We kindly request that you consider our revised submission for publication in Solid Earth. We would also like to extend our gratitude to the reviewers for their time, expertise, and dedication in helping us improve the quality of our work.

Thank you for considering our revised manuscript. We eagerly await your decision and remain at your disposal for any further information or clarification.

Sincerely,
Mattia de' Michieli Vitturi, Antonio Costa, Mauro A. Di Vito, Laura Sandri, Domenico M. Doronzo

**ANSWERS TO REVIEWERS**

Reviewers' comments are in bold.

RC1: 'Comment on egusphere-2023-1301', Anonymous Referee #1, 05 Sep 2023 reply

**COMMENT. Historically, flow thickness has been employed as a determinant factor. It might be instructive—though not imperative—to scrutinize the influence on dynamic pressure, particularly if a notable alteration is discerned. The manuscript is methodically robust and coherently structured. Both the objectives and the conclusions drawn are lucid and of significant import.**

ANSWER. *We thank the reviewer for the suggestion. We added a new figure in the paper showing two dynamic pressure maps. The figure is described in the paper by the following text:*

"Flow thickness may also be combined with dynamic pressure in order to assess, for different couples of thickness and dynamic pressure thresholds, the areas where these thresholds are exceeded simultaneously. Figure 13 shows, for two different thickness thresholds, the values of dynamic pressure exceeded during 24 hours of simulation. For example, in Figure 13b, the light green pixels represent the area where at some time the lahar produced, simultaneously, a thickness of at least 2 m and a dynamic pressure larger than 2000 Pa and smaller than 5000 Pa. "

[Figure]

*Figure 13.* Maps of exceedance of flow thickness and dynamic pressure: (a) thickness threshold 0.5m; (b) thickness threshold 2m. The colors represent the dynamic pressure thresholds exceeded during the 24 hours of simulation simultaneously with the thickness threshold.

**Technical corrections**

1. **Missing reference (citation in bibliography)**
   a. **Line 44 (IEA et al, 2016)**
   *The following reference has been added to the bibliography:*
   *International Atomic Energy Agency (IAEA): Volcanic Hazard Assessments for Nuclear Installations: Methods and Examples in Site Evaluation, IAEA-TECDOC-1795, IAEA, Vienna, 2016.*

      **b. Line 228 (Biagioli et al 2022) but bibliography says 2021**

*Fixed at line 228 (2021 is correct).*

      **c. Line 250 Is Sandri et al 2023 the same that Sandri et al (this issue)? If yes please change Sandri et al 2023 citation, if not reference is missing**

*Changed here and in other lines to Sandri et al (this issue)*

      **d. Line 334 missing reference (Bisson et al 2014)**

*The following reference has been added to the bibliography:*

*Bisson, M., Spinetti, C., & Sulpizio, R.: Volcaniclastic flow hazard zonation in the Sub-Apennine Vesuvian area using GIS and remote sensing. Geosphere, 10(6), 1419-1431, https://doi.org/10.1130/GES01041.1, 2014.*

      **e. Line 466 review reference (Hong at al, 2016) by (Hong et al.,2016)**

*Fixed*

  **2. Figures legend**

      **a. Figure 3 wrong legend (solid fraction title does not correspond with the**
          **i. caption and text)**

*The problem was with Figure 2 and not Figure3. Now the text and the caption has been fixed. Thanks!*

      **b. Please review Figure 5, the descriptions in the text (from line 277 to 304) do not correspond to the assigned label (a,b,c,d,e,f) figures that seem to be switched.**

*The reviewer is right. We switched the top and middle panels in the figure, and we fixed the text.*

      **c. Lines 443 are talking about Figure 8 but Figure 7 is referenced**

*Fixed*

      **d. Line 447 ¿Is it possible to show in Figure 7 the locations mentioned in line 447?**

*Added all the locations, apart from Acerra which is covered by the legend.*

      **e. Missing (a,b,c,d) in Figure 13**

Added letters and improved the quality of the figure.

      **f. Line 582 mention Figure 12a (does not exist)**

*We fixed the Figure reference (the correct one is 13a) and added a reference to Figure 16 in the text.*

      **g. Missing (a,b,c,d) in Figure 17**

Added letters and improved the quality of the figure.

**RC2**: 'Comment on egusphere-2023-1301', Gordon Woo, 03 Oct 2023 reply

**COMMENT. Using their enhanced modelling capability, it would be possible to consider alternative realisations of the 1631 events, inter alia, taking account of the substantial variability in the initial conditions and other model parameters. This counterfactual analysis would be very insightful and instructive for lahar risk analysis (Aspinall and Woo, 2019, Counterfactual analysis of runaway volcanic explosions, Front.Earth.Sci.). Such counterfactual risk analysis may be deferred to another publication, but some discussion would be useful for this paper.**

ANSWER. *We thank the reviewer for the suggestion. We added this sentence at the end of the paper:*

> *"The companion paper by Sandri et al. (this issue) will show an application of the presented model for hazard analysis of lahars from Vesuvius deposits in the Neapolitan area, where a wide range of initial conditions are investigated to produce probabilistic hazard maps To reach this goal, the companion paper considers eleven hydraulic catchments threatening the Campanian Plain, and in each catchment a large number of simulations accounts for the variability in the initial lahar volume, initial water fraction and initial mass load of the ashfall deposit. The database of simulations considered in the analysis by Sandri et al. (this issue) would allow one also to consider alternative realisations of the 1631 events, permitting for a counterfactual analysis that can be very insightful for lahar risk analysis (Aspinall and Woo, 2019), and it will be the focus of future research."*

*In addition, we added Aspinall and Woo (2019) to the list of references.*

**NEW REFERENCES ADDED TO THE MANUSCRIPT**

[revised manuscript text omitted]